# Fungal Selectivity and Biodegradation Effects by White and Brown Rot Fungi for Wood Biomass Pretreatment

**DOI:** 10.3390/polym15081957

**Published:** 2023-04-20

**Authors:** Jiyun Qi, Fangfang Li, Lu Jia, Xiaoyuan Zhang, Shuduan Deng, Bei Luo, Yonghui Zhou, Mizi Fan, Yan Xia

**Affiliations:** 1Yunnan Provincial Key Laboratory of Wood Adhesives and Glued Products, Southwest Forestry University, Kunming 650224, China; 2College of Engineering, Design and Physical Sciences, Brunel University London, Uxbridge UB8 3PH, UK

**Keywords:** fungi’s selectivity, biodegradation, softwood and hardwood, chemical components, biotechnology

## Abstract

The biodegradation path and mechanism of wood varies depending on diverse fungi and tree species, as fungi possess selectivity in degradation of versatile wood components. This paper aims to clarify the actual and precise selectivity of white and brown rot fungi and the biodegradation effects on different tree species. Softwood (*Pinus yunnanensis* and *Cunninghamia lanceolata*) and hardwood (*Populus yunnanensis* and *Hevea brasiliensis*) were subjected to a biopretreating process by white rot fungus *Trametes versicolor*, and brown rot fungi *Gloeophyllum trabeum* and *Rhodonia placenta* with various conversion periods. The results showed that the white rot fungus *Trametes versicolor* had a selective biodegradation in softwood, which preferentially convert wood hemicellulose and lignin, but cellulose was retained selectively. Conversely, *Trametes versicolor* achieved simultaneous conversion of cellulose, hemicellulose and lignin in hardwood. Both brown rot fungi species preferentially converted carbohydrates, but *R. placenta* had a selectivity for the conversion of cellulose. In addition, morphological observation showed that the microstructures within wood changed significantly, and the enlarged pores and the improved accessibility could be beneficial for the penetration and accessibility of treating substrates. The research outcomes could serve as fundamental knowhows and offer potentials for effective bioenergy production and bioengineering of bioresources, and provide a reference for further application of fungal biotechnology.

## 1. Introduction

Lignocellulosic biomass has been extensively used for the production of bioenergy and bioengineering over the past decades [1,2]. In a general production process of bioenergy or bioproduct, lignocellulosic biomass should be pretreated in order to modify their recalcitrant nature by reducing cellulose crystallinity and increasing the porosity. In the last few years, the microbial pretreatment process has become increasingly attractive as the pretreatment path of lignocellulosic biomass resources due to its environmentally benign process compared with traditional chemical pretreatment methods and its potential to reduce the inherent recalcitrance and enhance the enzymatic digestibility of the biomass [3,4,5]. A wide range of microorganisms, including bacteria as well as fungi, has been employed to pretreat cellulosic biomass, e.g., wood and non-wood, for bioenergy applications, among which rot fungi, such as white rot and brown rot fungi, received incremental attention as a cost-competitive alternative to traditional pretreatments [6,7,8]. However, different fungi could decompose different components of biomass by their own path through fungal infection and action mechanisms because of the extremely complicated interaction process between fungi and wood substrates, in which the chemical components and structures of the substrate, including the decay products, could affect and regulate their enzyme expression and activities and induce various biodegradation effects [9,10]. Besides, fungi are also capable of converting cellulose and lignin non-enzymatically by Fenton chemistry and make wood fibers brittle [11,12,13]. Therefore, the degradation effects of pretreatment by different fungi are critically different due to the various function paths and fungal selectivities.

Members of white rot fungi are able to decompose all structural components in lignocellulosic materials due to them having a complete extracellular enzyme system, which can generate not only hydrolytic enzymes (cellulase, hemicellulase, and pectinase), but also a wide variety of oxidative enzymes, such as Li-peroxidase, Mn-peroxidase and laccase. This endows them with the ability to efficiently convert lignin into H_2_O_2_ and CO_2_. On the other hand, white rot fungi convert lignocellulosic materials in a non-uniform path, in which white rot fungi can be classified into “selective” and “simultaneous” rot according to its degradation and removal of plant cell wall components; selective rot initially degrades hemicellulose and lignin, while cellulose is relatively retained whereassimultaneous rot degrades cellulose, hemicellulose and lignin in a rather uniform depletion [14,15,16]. Due to the unique function of white rot fungi in degrading lignin, the application in various aspects such as raw material pretreatment, biological bleaching, wastewater treatment and biological pulping has been emphasized, and significant results have been achieved in laboratory stage research. However, there are still problems with scaling-up the application of white rot fungi (such as low treatment efficiency, long pretreatment time for the long incubation period) contributed to by the various functions from different white rot fungi, whose information needs to be further improved and understood. Unlike the white rot fungi, brown rot fungi are powerful strains in degrading carbohydrates of biomass substrate but with little modification or degradation of lignin [17], causing rapid depolymerization of cellulose and hemicellulose due to the strong hydrolytic enzymes mainly secreted by brown rot fungi, which lack the ligninase secreting system. Except for the enzyme system, brown rot fungi can also operate Fenton reactions in carbohydrate decomposition and lignin modification [18,19]. It is worthy to note that the residual lignin after brown rot bioconversion can be useful for gasoline-based chemicals, biofuels, food additives and other biochemical materials. However, whether different kinds of brown rot fungi could decompose the cellulose and hemicellulose uniformly and/or selectively, particularly in different kinds of trees, is still not clear yet. In consequence, further research to achieve efficient and mild separation techniques for lignocellulosic feedstock but with no excessive degradation of substrates, and thereby improve the use value of residual substrate (cellulose or lignin), is a critical issue to be resolved. Therefore, the exploration on the fungal selectivity and the biodegradation effects by different kind of fungi will be favorable and beneficial for the potential application of fungal pretreatment in lignocellulosic bioconversion.

Wood has become one of the most important bioresources for potential biofuels and bioproducts, whether solid or liquid fuels, and the conversion is inevitably involved in both softwood and hardwood, which are different in chemical composition and micro-structure, and might lead to a difference in the bioconversion ability and mechanism of the decay fungi [20,21]. Various biodegradation paths may be deployed by fungi due to the difference in substrate structure and constituent components [22,23]. Fungi’s selectivity may be related to the species of both fungi and the inducer wood materials, especially so the latter [24]. To facilitate the effective conversion of wood as biomass during pretreatment for bioenergy and biochemical productions, a detailed knowledge about the fungi’s varying selectivity with tree species, different fungi and biopretreating stages is crucial and required.

This work aims to unveil the underlying white and brown rot fungi biodegradation mechanism of fast-growing tree species, the softwoods *Pinus yunnanensis* and *Cunninghamia lanceolata*, and the hardwoods *Populus yunnanensis* and *Hevea brasiliensis*, by comprehensively studying the chemical structure and composition variations during different decay stages through Fourier-transform infrared spectroscopy (FTIR) and X-ray photoelectronic spectroscopy (XPS) analysis. Crystalline structure and microstructure were also characterized to explore the structure-performance relationship, contributing to the establishment of the information of fungal selectivity and the biodegradation effects on different kinds of tree species decayed by different fungi. In addition to achieve a thorough understanding of fungi’s intrinsic motivation and selectivity and the biodegradation path among wood species, this current study could provide a great potential and feasible prospect for the efficient bioconversion and biotechnology application through proper fungi selection, including the combined fungal approach in futural treatment for shortening time.

## 2. Materials and Methods

### 2.1. Materials

Small wood blocks (20 mm (R) × 20 mm (T) × 10 mm (L)) without any defects, such as resin, gum or knots and decay, were prepared from softwood (*Pinus yunnanensis*, *Cunninghamia lanceolata*), and hardwood (*Populus yunnanensis*, *Hevea brasiliensis*). White rot fungus *Trametes versicolor* (L.) Lloyd. (*T. versicolor*), and brown rot fungi *Gloeophyllum trabeum* (Pers.: Fr.) Murrill (*G. trabeum*) and *Rhodonia placenta* (Fr.) Niemelä, K.H.Larss. & Schigel (*R. placenta*) were obtained from China Forestry Culture Collection Centre.

### 2.2. Biodegradation Decay Assessment

The laboratory biodegradation was carried out according to the Chinese Standard GB/T13942.1–2009. *T. versicolor*, *G. trabeum* and *R. placenta* were incubated at 28 °C and 85% relative humidity, respectively, in an incubator until mycelium covered the entire surface of the agar. Subsequently, the sterilized wood block samples from the autoclave were transferred to the cultures of fungi, 6 samples for each stage (4, 8, 12, 16 weeks) for every kind of tree species and fungi group. After fungi exposure, the samples were removed from the incubator and fungal mycelium were removed from the surface of the specimens and the mass losses of wood samples were calculated based on the oven-dried weight.

### 2.3. FTIR Analysis

The wood samples were ground into 40–60 mesh wood flour, then the wood powder was examined for the analysis of chemical characterization by FTIR-650 equipment. Potassium bromide (KBr) was used to collect the background. The dried wood powder was mixed with KBr at a weight ratio of 1:100, then pressed into a sheet. The FTIR spectra of all samples were determined to be in the range of 500–4000 cm^−1^. Samples were examined at a spectral resolution of 4 cm^−1^ with 32 scans per sample.

### 2.4. X-ray Diffraction Analysis

The dried wood block samples were pulverized into a fine powder (60–80 mesh screen) and characterized with X-ray diffraction (XRD) using UItima IV 40 kV and 40 mA at an angle of scanning of 5–40° (2θ). The scanning rate was 0.4 s/step with a total of 2685 steps.

### 2.5. XPS Spectroscopy Analysis

XPS analysis was carried out by Thermo Scientific K-Alpha+ X-ray Photoelectron Spectrometer equipped with a monochromatic AlKa X-ray Source (14,876.6 eV). Photoelectrons were detected with a 180° double focusing hemispherical analyzer operated in the constant analyzer energy (CAE) mode. A base pressure of 5 × 10^−9^ mbar was held in the analytical chamber. X-ray beam intensity was adjusted as designed by changing the electron current striking the anode (15 mA), while the acceleration voltage was held at 15 kV. High-resolution spectrum of the C1s region from 280 to 300 eV was collected.

### 2.6. Observation on Microscopic Structure

Nikon 80i biological digital microscope, scanning electron microscope (SEM, Hitachi TM3000, Japan) and the Confocal Raman Microscopy (CRM, LabRam Xplora, HORIBA) were used to observe the mycelial distribution inside the wood cell and analyze the microscopic structural changes of samples.

## 3. Results and Discussion

### 3.1. Wood Mass Conversion in Relation to Fungi and Wood Species

Mass conversion was examined by using the mass losses of wood samples in different biodegradation periods and are summarized in Table 1. It can be seen that the mass conversion of softwoods decayed by both two brown rot fungi was higher than that by white rot fungus *Trametes versicolor*, with the mass losses of 28.59% and 36.19% decayed by *G. trabeum* and *R. placenta* compared to 13.09% decayed by *T. versicolor* in *Pinus yunnanensis* wood, and 66.52% and 45.87% decayed by *G. trabeum* and *R. placenta,* respectively, compared to 35.57% decayed by *T. versicolor* in *Cunninghamia lanceolata* wood. In contrast, the mass conversion of hardwoods decayed by white rot fungus exceeded that by brown rot fungi, where the mass loss was 40.75% decayed by *T. versicolor* compared to 32.65% and 19.05% decayed by *G. trabeum* and *R. placenta* in *Hevea brasiliensis*, as well as the mass loss of80.88% when decayed by *T. versicolor* compared to 65.54% and 36.88% decayed by *G. trabeum* and *R. placenta* in *Populus yunnanensis* wood. This discrepancy could be ascribed to the different types of lignin, in which there are guaiacyl lignin and syringyl lignin, and as such more methoxyl groups in hardwood were more easily degraded [25,26,27]. Therefore, softwood could possess more resistance against white rot fungi, which also decay lignin, as the softwood only contains guaiacyl lignin type and more lignin content as given in Table 2, which was compiled from the literature. Therefore, the brown rot fungi were clearly more vigorous and aggressive than white rot fungi in converting softwood species, opposed to the stronger degradation ability of white rot fungi in converting hardwood than that of two brown rot fungi. Collectively, it could be suggested that there were different biodegradation paths between soft and hard wood, contributed by different chemical compositions between softwoods and hardwoods [28,29,30,31,32,33].

Furthermore, it was worth mentioning that *Pinus yunnanensis* was more resistant to conversion than *Cunninghamia lanceolata* in the softwood group, and *Hevea brasiliensis* also exhibited more resistance than *Populus yunnanensis* in the hardwood group due to the noticeable variety in the densities from the wood tree species [34,35], as shown in Table 3, in which the densities of *Pinus yunnanensis* and *Hevea brasiliensis* were higher than *Cunninghamia lanceolata* and *Populus yunnanensis,* respectively, with the densities being 0.472 g/cm^3^ and 0.401 g/cm^3^ in softwood group, and 0.650 g/cm^3^ and 0.364 g/cm^3^ in hardwood group, respectively. This indicates that the density is a factor, influencing the conversion of wood subjected to decay fungi, as the loose fiber structure means that the cell micropores were large to be penetrated and accordingly eroded by fungi more easily [36,37].

### 3.2. Chemical Variation of Wood Biomass after Selective Fungi Process

To analyze the changes of chemical characterization, the FT-IR spectra of the samples are presented in Figure 1. In the softwood group, when samples were exposed to the white rot fungus *T. versicolor*, the absorption peaks at 1735 cm^−1^, 1510 cm^−1^ and 1266 cm^−1^ shifted to lower wavelengths before 8 weeks of decay; manifesting *T. versicolor* initially degraded hemicellulose and lignin, while cellulose was retained, which caused selective rot. The reason for the selective rot of softwood was that the tough S_3_ layer of the tracheid cell wall prevented the mycelium developing outward from the lumen, as the concentration of lignin in S_3_ was much lower than other regions of the cell wall structure, leading to preferential depolymerization of lignin [41,42]. In brown rot fungi, the bands at 1735 cm^−1^ and 1042 cm^−1^ decreased at 8 weeks, indicating that polysaccharides were preferentially degraded. This could be caused by the powerful capacity of secreting hydrolytic enzymes (cellulase and hemicellulase) by the brown rot fungi, which also is in line with the previous publishments that brown rot fungi depolymerized wood by selectively degrading hemicellulose and cellulose, and hemicelluloses are destroyed faster than cellulose [43].

In hardwood samples, the absorption peak at 1735 cm^−1^ was reduced at 4 weeks decay, the bonds at 1510 cm^−1^, 1245 cm^−1^ and 1045 cm^−1^ were decreased after *T. versicolor* decayed for 8 weeks, showing hemicellulose degraded preferentially in the early stage, followed by lignin and cellulose, which caused simultaneous rot in both hardwood samples. This could explain why the branched chain structure of hemicellulose was more easily eroded by fungi in the initial stage, which might lead to the breakup of lignin-carbohydrate complex bonds that link hemicellulose and lignin [44,45], contributing to the easy structural destruction of lignin with the prolonged pretreating stage. In two brown rot fungi, the absorption peaks at 1735 cm^−1^, 1400 cm^−1^ and 1042 cm^−1^ were reduced at 4 weeks decay, the bands at 1626 cm^−1^ and 1245 cm^−1^ decreased after 8 weeks decay, suggesting that fungi could degrade the polysaccharide initially, followed by the lignin in the advanced stage [46,47].

Results suggested that white rot fungus *Trametes versicolor* could attack lignin selectively and cause a selective conversion in softwood *Pinus yunnanensis* and *Cunninghamia lanceolata*, which could enhance the accessibility of cellulose to enzymatic digestibility, whereas it caused a simultaneous conversion in the hardwoods *Populus yunnanensis* and *Hevea brasiliensis*.

### 3.3. Change in Crystallinity

The crystallinities of woods after fungi exposure are determined by XRD and the results are depicted in Figure 2, which can also be corresponded to the mass losses (Table 1). An obvious trend was observed that the crystallinities of woods exposed to the white rot fungus *T. versicolor* elevated during the decay stage, with the crystallinities increasing by 11.17% in *Pinus yunnanensis* and 16.90% in *Populus yunnanensis*. The main reason is that the amorphous structures of hemicellulose and lignin, especially the hemicellulose with many branched chains, could be easily eroded and deconstructed by white rot fungi, which can secrete the oxidase enzymes, such as laccase, lignin peroxidase (Li-peroxidase) and manganese peroxidase (Mn-Peroxidase), leaving the relatively higher concentration of structured cellulose and hence resulting in relatively increased crystallinity [45].

Both brown rot fungi exposures resulted in the continuous decrease in the crystallinities of wood samples during the conversion period. As only cellulose has a crystalline structure among the three main chemical compositions of wood, this trend obtained from crystallinities suggests that cellulose could be depolymerized and converted more severely compared with hemicellulose and lignin [48,49]. Notably, the crystallinities of *Hevea brasiliensis* and *Populus yunnanensis* exposed to *R. placenta* decreased from 30.26% and 22.97% to 28.25% and 18.49% at 4 weeks respectively, while the mass loss was only 0.08% and 0.38%. This phenomenon implied that *R. placenta* had selectivity for deconstruction in cellulose, and it was in the reproducing and growing state in the initial stage, but they could already impact and even decompose cellulose macromolecules, contributing to the decrease in the degree of polymerization [50,51].

### 3.4. XPS Spectroscopy Analysis

In order to obtain the surface chemical composition, XPS analysis was carried out to investigate the changes of carbon atoms C1–C3, and results are portrayed in Figure 3 and Table 4. It can be seen that for the white rot group, whether softwood or hardwood, the ratio of C1 decreased, which mainly came from lignin and wood extract, while the ratio of C2 increased, which has been proved to be mainly from cellulose due to a carbon bound to a single noncarbonyl oxygen atom [52]. For the brown rot groups, the results were in an opposite situation. The reason is attributed to that white rot fungi can secret lignin enzymes, such as laccase, Mn-peroxidase and Li-peroxidase to degrade lignin, while brown rot fungi can mainly secret hydrolases but lack the ability to secrete lignin enzymes, contributing to the selective degradation of carbohydrates [52,53,54,55,56,57]. In addition, the ratio of C2 decreased more obviously in *R. placenta* than in *G. trabeum*, and it also proved that *R. placenta* could decompose cellulose selectively, which is in agreement with the results in XRD analysis. This path of *R. placenta* is different with brown rot fungus *G. trabeum* which stated in previous research that *G. trabeum* had a tendency to convert hemicellulose [50]. Therefore, it can be inferred that these two kinds of brown rot fungi have their own degradation path and different selectivity for carbohydrates (cellulose and hemicellulose), which can be applied for biotechnology in different fields.

### 3.5. Mycelial Distribution and Conversion of Constituents

The microstructural changes in wood cell walls and the mycelial distribution observed through fluorescence microscope and SEM showed that at the beginning, mycelium appeared in the cell lumina and spread into fibers as the time increased for all three fungi (Figure 4 and Figure 5), and then invaded the membrane of pits and damaged the cell walls. In white rot fungus *T. versicolor* group, it can be observed that fungal strains already colonized in wood structures only at 4 weeks due to the obvious hyphae in cell lumina, then hyphae appeared in clusters after 8 weeks. Furthermore, the fluorescence intensity dimmed compared with the sound wood as well. It can be inferred that *T. versicolor* depolymerized the lignin along with the colonization of the hyphae and spread in the cell lumina [58]. After exposed to brown rot fungi, the fluorescence intensity of wood samples weakened and the cell walls became thinner at 16 weeks, implying lignin can be modified at the advanced stages [56]. In addition, a large number of pits in cross fields were destroyed, indicating fungi were more likely to grow and reproduce in the cross fields, resulting in the destruction of the pits. Furthermore, it was found that fungi attacked to parenchyma cells via pits and the wood rays were the primary paths for the spread of mycelium [57].

To understand the underlying mechanism of the chemical compositions transitions in wood cell walls more clearly, the technique of Confocal Raman Microscopy was performed in this current work. Confocal Raman Microscopy can provide an intuitionistic observation of lignin and carbohydrates in cell walls with a high resolution, which is crucial to the understanding of the interaction mechanism between fungi and wood components. As shown in Figure 6 and Figure 7, in wood samples converted by white rot fungus *T. versicolor*, the distribution concentration of lignin reduced severely, while carbohydrates were retained. On the contrary, both brown rot fungi exposures resulted in obvious and extensive deconstruction of carbohydrates in cell walls due to the relatively low content of carbohydrates distribution, while the lignin concentrations were relatively increased, indicating that brown rot fungi had a preference for the bioconversion of carbohydrates.

Microstructure observation showed that the hyphae of decay fungi colonized and spread in the lumina of the wood cells, and caused the greater disruption of the integrity within the cell walls, which created large pores and could facilitate the enzymatic digestion due to the increased accessibility [19,59]. The constituent components of wood could be selectively depolymerized with the disruption of the wood tissues as well.

## 4. Conclusions

Both white and brown rot fungi’s actual selectivity and biodegradation effects on softwood and hardwood (including four tree species) have been investigated, through analyzing the accompanied changes of the chemical components and the microstructural features of cell walls during various bioconversion stages. In softwood, white-rot fungus *T. versicolor* could degrade lignin and hemicelluloses extensively and reduced the inherent recalcitrance of biomass with a limited degradation of cellulose. Conversely, *T. versicolor* caused simultaneous conversion in hardwood, which achieved the uniform conversion of cellulose, hemicellulose and lignin. Brown rot fungi caused selective deconstruction of the carbohydrate polymers, namely cellulose and hemicellulose of wood cell walls, and left most of the lignin unattacked, especially the *R. placenta*, reducing the crystallinity of cellulose and enhancing the enzymatic digestibility. Therefore, the fungal pretreatment increased the porosity and improved the accessibility of wood cells. In consequence, the *T. versicolor* pretreatment could be beneficial for maximizing the cellulose content of softwood, which is favorable in biological pulping or bioethanol industries, whereas brown rot fungal pretreatment could be profitable for gasoline-based chemicals, biofuels and other biochemical materials, particularly the *R. placenta*. The outcomes of the study indicate that the fast-growing tree species could provide enormous potential for effective bioconversion with the designed fungi conversion, as well as offer a promising prospect for optimal bioengineering and biotechnology of value added bioproducts through proper fungi selection including the combined fungal treatments.

## Figures and Tables

**Figure 1 polymers-15-01957-f001:**
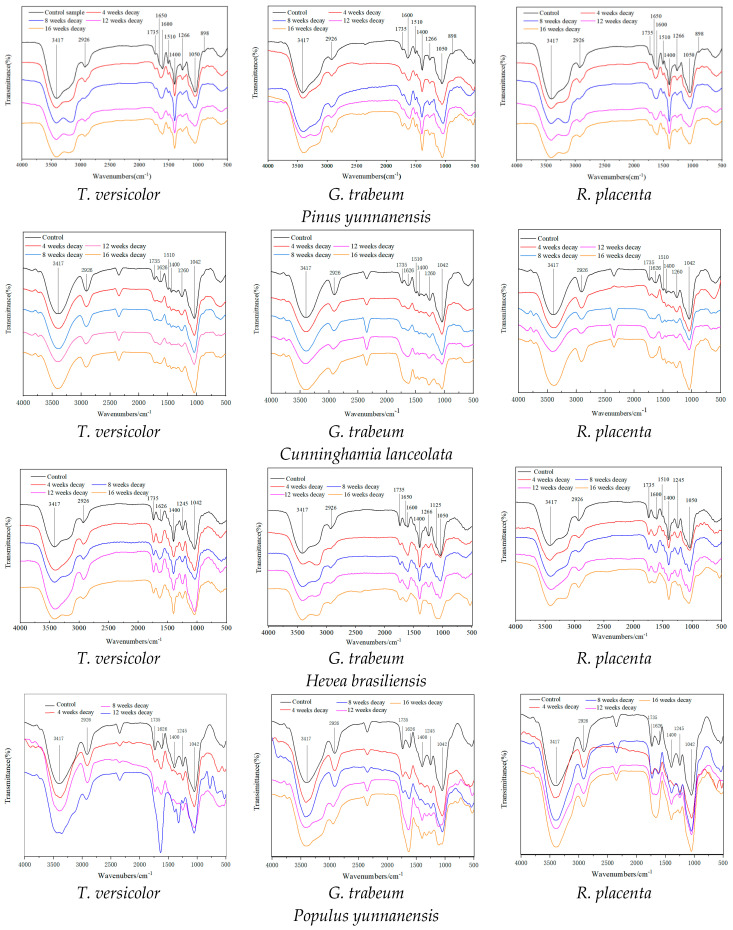
FT−IR spectra of wood samples biodegraded by different fungi.

**Figure 2 polymers-15-01957-f002:**
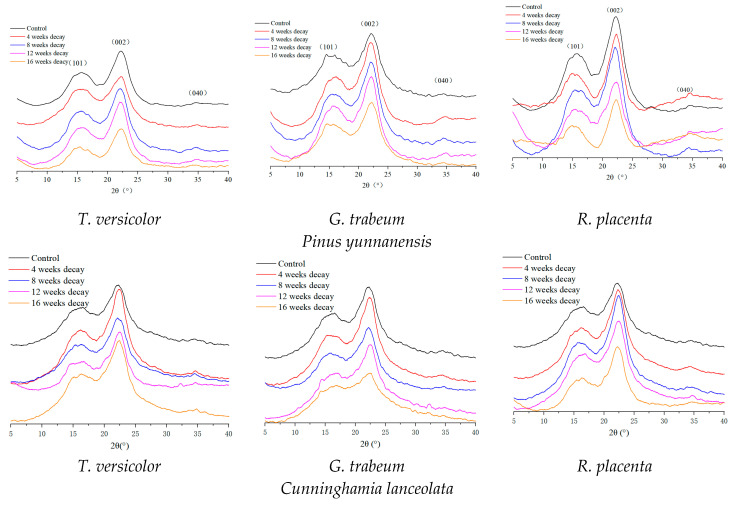
XRD patterns of wood samples converted by different fungi.

**Figure 3 polymers-15-01957-f003:**
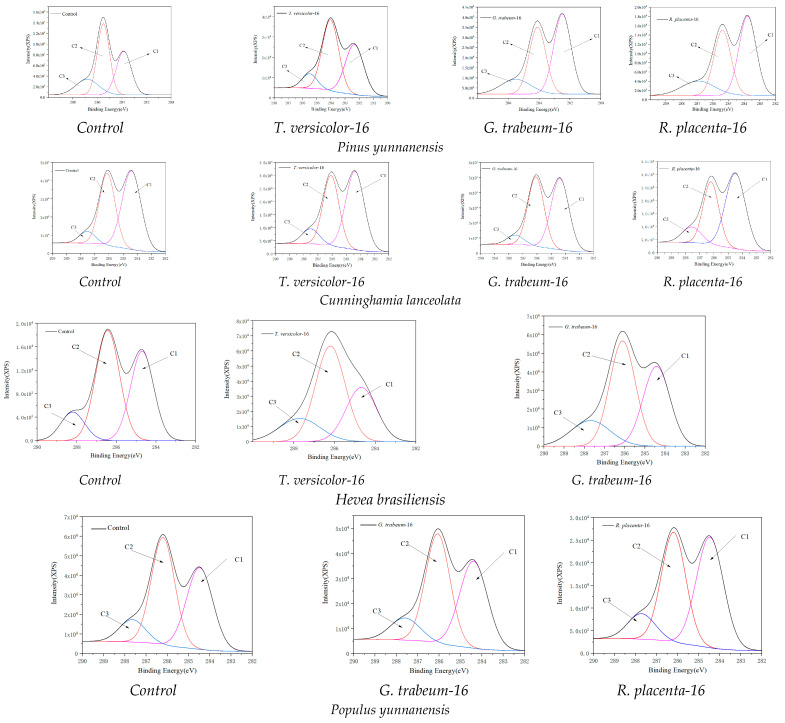
XPS scan of C1s region of wood samples converted by different fungi.

**Figure 4 polymers-15-01957-f004:**
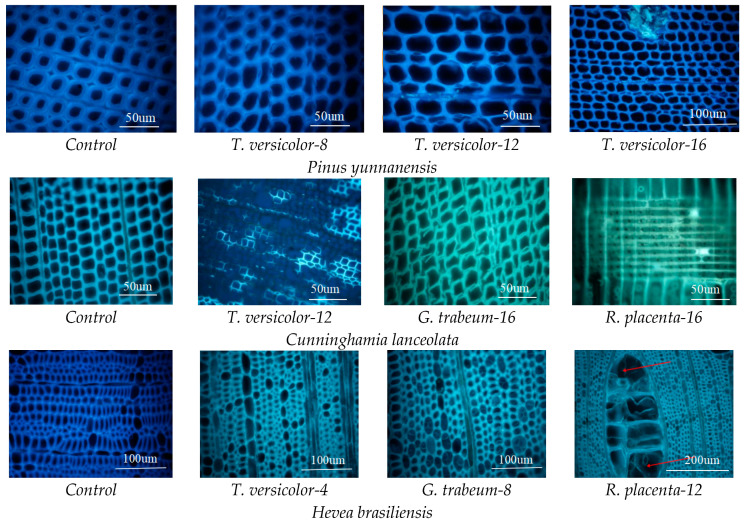
Light microscopy images of wood samples biodegraded by different fungi.

**Figure 5 polymers-15-01957-f005:**
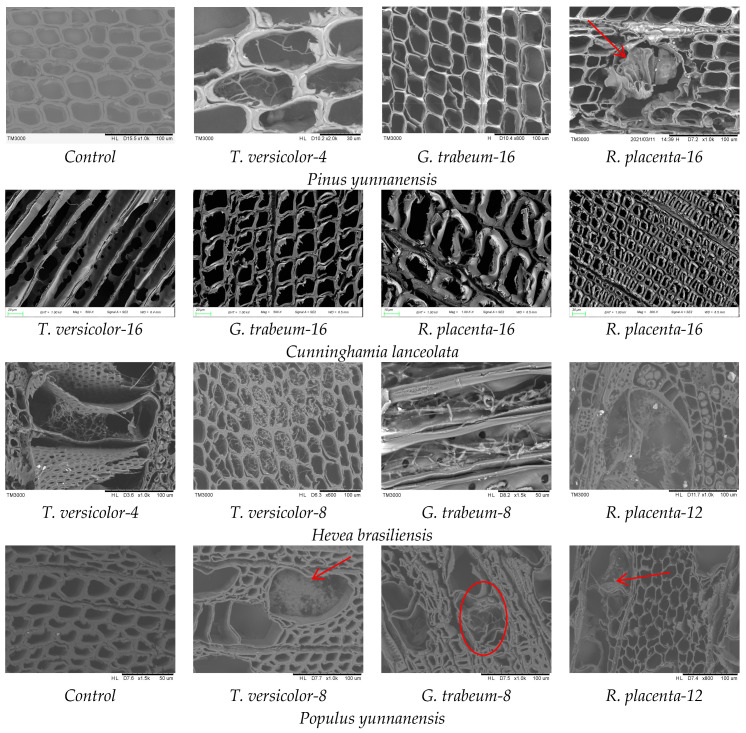
SEM images of wood samples biodegraded by different fungi.

**Figure 6 polymers-15-01957-f006:**
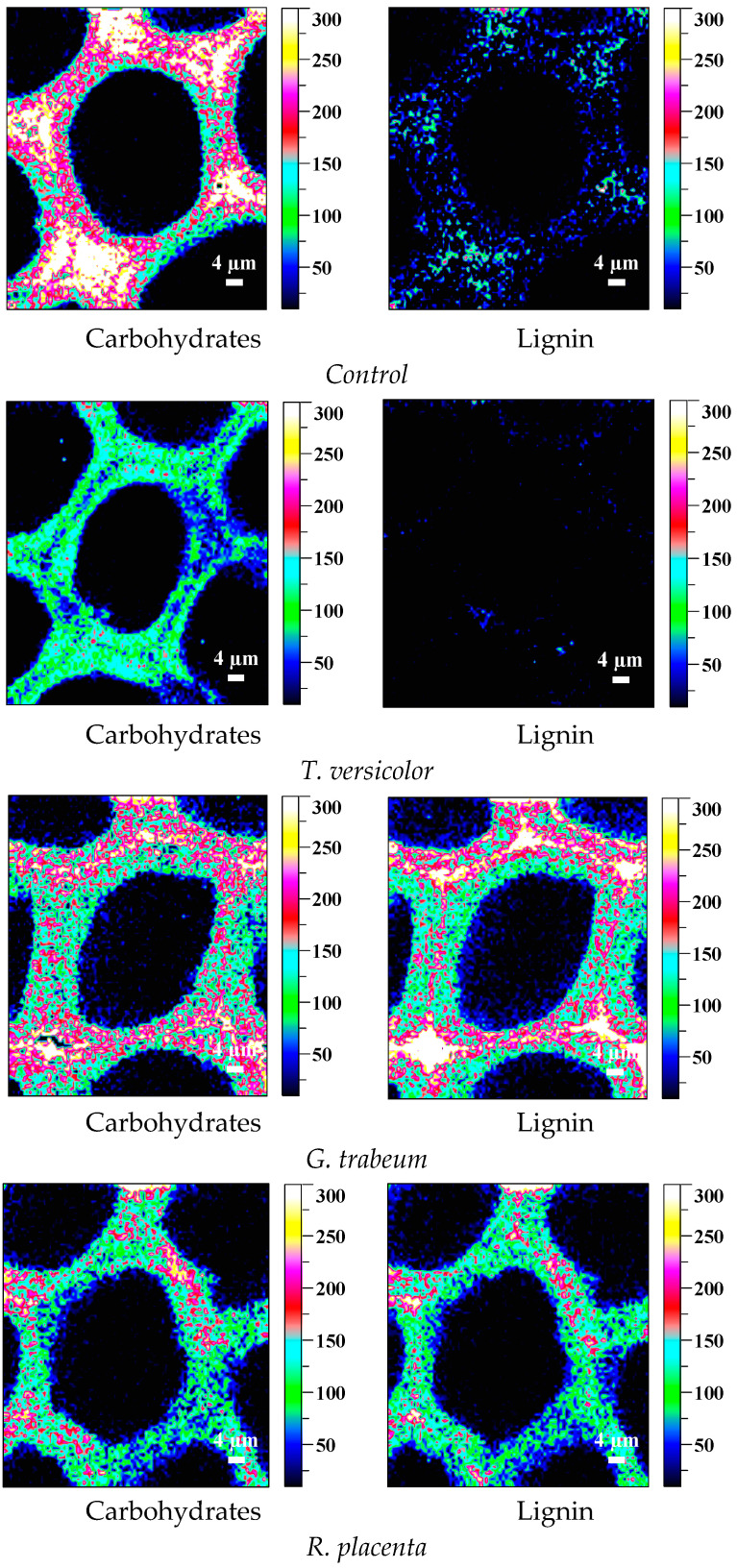
Raman images of *Pinus yunnanensis* samples converted by different fungi for 16 weeks.

**Figure 7 polymers-15-01957-f007:**
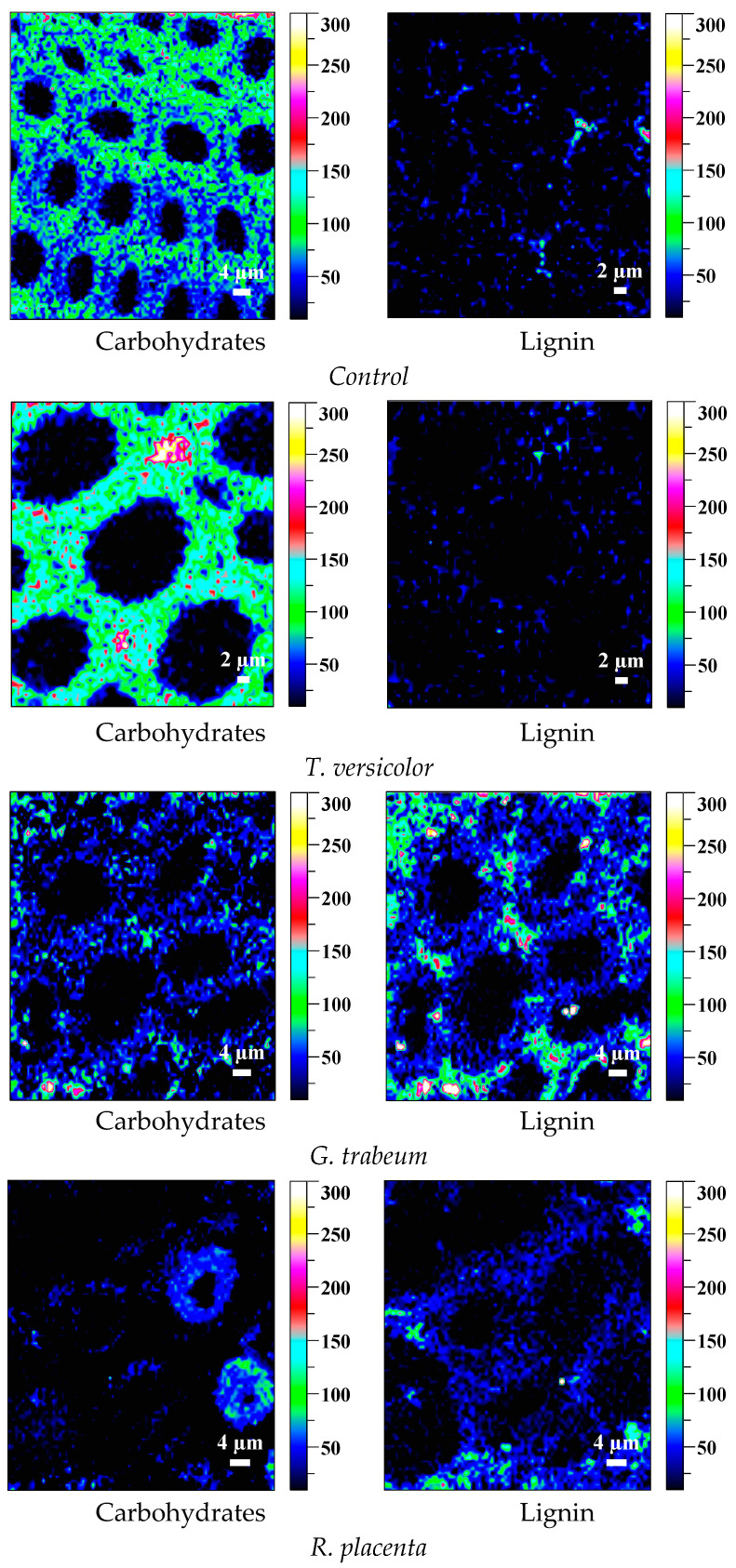
Raman images of *Hevea brasiliensis* samples converted by different fungi for 16 weeks.

**Table 1 polymers-15-01957-t001:** Mass conversions and crystallinities of wood samples at different stages.

	Tree Species	Periods (Weeks)	Mass Loss/%	Standard Deviation (б)	CrI/%	Varying Ratio of CrI/%	Periods (Weeks)	Mass Loss/%	Standard Deviation (б)	CrI/%	Varying Ratio of CrI/%
Fungi		*Pinus yunnanensis*					*Cunninghamia lanceolata*				
*T versicolor*		Control	0	0	32.82	0	Control	0	0	22.30	0
		4	1.02	0.2927	20.01	−12.81	4	0.17	0.5436	22.42	+0.12
		8	1.41	0.3444	24.04	−8.78	8	1.99	0.3649	22.53	+0.23
		12	10.59	0.8408	28.24	−4.58	12	31.18	0.6254	26.45	+4.15
		16	13.09	0.6817	43.99	+11.17	16	35.57	0.3583	22.96	+0.66
*G. trabeum*		4	0.98	0.5464	27.78	−5.04	4	0.60	0.4528	20.00	−2.30
		8	5.10	0.4988	25.67	−7.15	8	3.98	0.3745	18.46	−3.84
		12	13.03	0.5297	24.52	−8.30	12	51.36	0.6283	17.46	−4.84
		16	28.59	0.3244	22.05	−10.77	16	66.52	0.3683	16.42	−5.88
*R. placenta*		4	7.71	0.5197	19.80	−13.02	4	0.50	0.3659	19.78	−2.52
		8	7.36	0.4225	16.66	−16.16	8	1.71	0.4573	19.94	−2.36
		12	35.21	0.2407	15.37	−17.45	12	38.05	0.6538	18.91	−3.39
		16	36.19	0.3513	17.11	−15.71	16	45.87	0.5527	18.83	−3.47
		*Hevea brasiliensis*					*Populus yunnanensis*				
*T versicolor*		Control	0	0	30.26	0	Control	0	0	22.97	0
		4	4.97	0.5593	36.71	+6.45	4	4.34	0.6366	22.25	−0.72
		8	5.57	0.4917	35.69	+5.43	8	6.56	0.2846	24.37	+1.40
		12	19.22	0.4489	30.87	+0.61	12	80.88	0.7354	39.87	+16.90
		16	40.75	0.2305	26.15	−4.11	16	/	/	/	/
*G. trabeum*		4	2.03	0.3068	30.15	−0.11	4	0.25	0.4683	20.83	−2.14
		8	5.61	0.2172	26.45	−3.81	8	3.48	0.5739	20.55	−2.42
		12	27.70	0.6643	29.94	−0.32	12	61.20	0.2648	15.24	−7.73
		16	32.65	0.2370	18.37	−11.89	16	65.54	0.8464	18.17	−4.80
*R. placenta*		4	0.08	0.0353	28.25	−2.01	4	0.38	0.5628	18.49	−4.48
		8	2.02	0.3779	26.69	−3.57	8	0.90	0.6783	17.77	−5.20
		12	10.25	0.5247	16.64	−13.62	12	16.46	0.5728	17.46	−5.51
		16	19.05	0.4543	14.06	−16.20	16	36.88	0.3558	17.34	−5.63

**Table 2 polymers-15-01957-t002:** The average of relative chemical compositions of different tree species.

	Cellulose (%)	Hemicellulose (%)	Lignin (%)
*Pinus yunnanensis*	49.65	16.20	26.41
*Cunninghamia lanceolata*	44.39	11.91	26.76
*Hevea brasiliensis*	50.19	24.31	23.43
*Populus yunnanensis*	50.22	14.55	24.59

The relative chemical compositions of different tree species refering to previous publications [23,24,25,26].

**Table 3 polymers-15-01957-t003:** Densities of different tree species.

Tree Species	*Pinus yunnanensis*	*Cunninghamia lanceolata*	*Hevea brasiliensis*	*Populus yunnanensis*
Densities/g.cm^−3^	0.472	0.401	0.650	0.364
Standard deviation (б)	0.0061	0.0197	0.0100	0.0355

The densities of different tree species refer to previous publications [37,38,39,40].

**Table 4 polymers-15-01957-t004:** C (1s) Division peak percentage contents of wood surface after biodegradation.

Tree Species	Fungi	Periods (Weeks)	C1/C	C2/C	C3/C
	Control	0	34.81	49.93	15.25
*Pinus yunnanensis*	*T versicolor*	16	33.24	53.17	13.59
	*G. trabeum*	16	47.56	40.11	12.33
	*R. placenta*	16	58.60	29.50	11.89
	Control	0	53.05	40.77	6.18
*Cunninghamia lanceolata*	*T versicolor*	16	50.19	43.69	6.13
	*G. trabeum*	16	53.89	35.43	10.66
	*R. placenta*	16	51.53	39.62	9.08
	Control	0	37.69	52.23	10.09
*Hevea brasiliensis*	*T versicolor*	16	30.58	51.95	17.47
	*G. trabeum*	16	36.67	46.15	17.17
	Control	0	35.82	55.70	8.48
*Populus yunnanensis*	*G. trabeum*	16	41.99	46.82	11.19
	*R. placenta*	16	46.67	42.10	11.23

## Data Availability

The data presented in this study are available from the listed authors.

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
