# Peer review of "Fungal Selectivity and Biodegradation Effects by White and Brown Rot Fungi for Wood Biomass Pretreatment"

_polymers, 2023, doi:10.3390/polym15081957_

Round 1

Reviewer 1 Report

The article presents information on fungal selectivity and biodegradation by white and brown rot fungi for wood biomass pretreatment. However, extended information on this matter already exists since 1985: white rot fungi are able to degrade the lignin, cellulose, and hemicellulose components of wood to carbon dioxide (Kirk and Shimada 1985). In contrast, brown rot fungi mainly degrade the cellulose and hemicellulose, but only slightly modify the lignin (Kirk and Moore 1972; Kirk and Highley 1973); Witomski, P., Olek, W., and Bonarski, J. T. (2014). "Effects of white and brown rot decay on changes of wood ultrastructure," BioRes. 9(4), 7363-7371; 10.1016/j.enconman.2017.03.021; etc.

Some sentences are found also in  https://doi.org/10.1016/j.indcrop.2022.115726 

1. Please clearly state the novelty of this study. Why is this important? What are the limitations of your study?

2. Some information on the enzymes involved in biodegradation during the fungal pretreatment and the operating parameters governing performance of the fungal pretreatment would increase the quality of the article

3. Analysis of white rot-degraded wood indicates that lignin degradation by these microorganisms is highly oxidative and may involve chemical oxidants such as singlet oxygen and hydroxyl radicals. Was this investigated?

4. The authors did not respect the template:

-there is no affiliation mentioned

-the highlights are presented in the main document

- pg 2 begins with a picture (graphical abstract I presume) without any relations with the manuscript or Legend

5. Microorganisms are not written in italic in the first part of the Introduction

6. The Raman images are too large

7. The title is not accurate: the biodegradation path is not presented as it should imply the enzymes involved and the chemical components for each step. More suitable title should be selected for the article.

8. It is suggested to compare the results of the present research with some similar studies which is done before.

9.  Please make sure your conclusions' section underscore the scientific value added of your paper, and/or the applicability of your findings/results. Please revise your conclusion part into more details. Basically, you should enhance your contributions, limitations, underscore the scientific value added of your paper, and/or the applicability of your findings/results and future study in this session.

Reviewer 2 Report

Dear Editors and authors,

this research work is good yet must be further improved. Actually there are some really good and valuable parts in it (presentation of graphs, used analytical methods, figures) but there are also many shortcommings.

First of all English language is poor and must be improved. It is so bad that it also affects readability and makes text abmibious at some places. There are so many errors that there is no use to list them. In short: Please let it checked by a native English speaker before next submission.

Graphical abstract is difficult to interpret what is actually ment in the figure.

It is not understood why the use of nomenclature: you refer to Pinus yunnanensis as "Pinus" but in case of Cunninghamia Lancelota "C. lancelota".. Using "Pinus" and "Populus" as sample names is ambigious, please make this uniform.

L119: spectrum collections is not proper here use spectrum acquisition

L124: what is 60-80 mesh?

L151: respectively? This is also a language issue but it is badly formatted here. What the main question here is: are the differences statistically significant OR not? Was there any test conducted on statistics? There is no standrd deviation of data included in table 1. Please included standard deviation for the measurements and indicate if values differ significantly at what p level.

L176-177. What pioneers?? This sentence is ambigious.

Table3: I believe that these values are from literature (aka. from the work "pioneers") but wood density varies also within species please indicate if applicable some range within species (eg using standard deviation or confidence range)

L235: please use hardwood and softwood terms consequently.

L245: not "oxidase" but use "oxidase enzymes"

L262: "seriously" seriously?

Table4: are there any standard deviation data available for these values? Are the differences significant?

L346: "lignin distribution concentration was severe". What do you mean? Do you mean the concentration , the distribution of the lingnin content and what precisely does "severe" mean?

Fig 6. The tags COntrol, T. versicolor, G. trabeum and R. placenta are placed awkwardly they are much closer to the underlying figures and this is disturbing, please align these tags properly. Same applies to Fig 7.

Round 2

Reviewer 1 Report

The article can be published.

Reviewer 2 Report

The authors have altered the manuscript according to the suggestions and can be accepted. However, please make a final check for typos and English language.